

# MINTIA: a metagenomic INserT integrated assembly and annotation tool

Philippe Bardou[1,*], Sandrine Laguerre[2,*], Sarah Maman Haddad[1], Sabrina Legoueix Rodriguez[3], Elisabeth Laville[2], Claire Dumon[2], Gabrielle Potocki-Veronese[2] and Christophe Klopp[4]

[1] Sigenae, GenPhySE, Université de Toulouse, INRAE, ENVT, Castanet Tolosan, France
[2] TBI, Université de Toulouse, CNRS, INRAE, INSA, Toulouse, France
[3] TWB, Universite de Toulouse, INRAE, INSA, CNRS, Ramonville-Saint-Agne, France
[4] Sigenae, Genotoul Bioinfo, MIAT UR875, INRAE, Castanet Tolosan, France
[*] These authors contributed equally to this work.

## ABSTRACT

The earth harbors trillions of bacterial species adapted to very diverse ecosystems thanks to specific metabolic function acquisition. Most of the genes responsible for these functions belong to uncultured bacteria and are still to be discovered. Functional metagenomics based on activity screening is a classical way to retrieve these genes from microbiomes. This approach is based on the insertion of large metagenomic DNA fragments into a vector and transformation of a host to express heterologous genes. Metagenomic libraries are then screened for activities of interest, and the metagenomic DNA inserts of active clones are extracted to be sequenced and analysed to identify genes that are responsible for the detected activity. Hundreds of metagenomics sequences found using this strategy have already been published in public databases. Here we present the MINTIA software package enabling biologists to easily generate and analyze large metagenomic sequence sets, retrieved after activity-based screening. It filters reads, performs assembly, removes cloning vector, annotates open reading frames and generates user friendly reports as well as files ready for submission to international sequence repositories. The software package can be downloaded from https://github.com/Bios4Biol/MINTIA.

# INTRODUCTION

Microbial ecosystems are unrivaled sources of new protein functions. Thanks to advances in sequencing technologies, microbial ecosystems have been largely explored during the last decades. In the same time, various software packages and analytic pipelines have been developed to assemble, annotate and explore massive sequencing data sets. Everyday, sequence databases are enriched with new metagenomic sequences revealing the tremendous amount of putative functions that can be found in the nonculturable species of microbial ecosystems. However, 20 to 50% of putative genes remain unannotated due to too few homologies with previously characterized proteins, which represents

Corresponding author
Philippe Bardou,
philippe.bardou@inrae.fr

only 2% of putative genes in the trEMBL/RefSeq databases (*Stein, 2001*; *Aubourg & Rouze, 2001*; *Pruitt, 2004*; *Stothard & Wishart, 2006*; *Harrington et al., 2007*; *Gilbert et al., 2010*). In order to explore complex ecosystem functional diversity and characterize it at the molecular level, activity-based metagenomics represents the most powerful *in vitro* technology. Its throughput, cost and versatility have been continuously improved in the last decade, in particular thanks to droplet microfluidics (*Tauzin et al., 2020*). The activity-based metagenomics approach includes four steps: (i) inserting DNA fragments extracted from an environmental sample in an expression vector (cosmids, fosmids or bacterial artificial chromosomes), (ii) transforming a microbial expression host to create a metagenomic library, (iii) screening the clone phenotype using selective media, chromogenic/fluorogenic substrates or reporter systems to isolate the hit clones producing the targeted activity, and last, (iv) sequencing multiplexed metagenomic inserts of the hit clones using NGS technologies either after an individual DNA barcoding step (*Tasse et al., 2010*) or directly, without marking them (*Lam et al., 2014*), (v) obtaining DNA sequences in order to identify genes responsible for the screened activity (*Healy et al., 1995*). Using this approach, a protein function can be assessed without any prior information on its sequence. Genes, their genomic context and taxonomical origin are determined in order to obtain new insights into the relationships between gene functions, bacterial taxa, microbiome prevalence and contribution to ecosystems. Finally, it rationalizes sequencing efforts by only focusing on clones exhibiting the targeted activity. A careful analysis of large metagenomic DNA inserts, comprising entire metabolic pathways, is a critical step for functional studies. It requests suitable bioinformatic resources to assemble reads into contigs, remove vectors, detect coding sequences (CDS), translate, align and compare them to reference databases. To perform these different steps, freely available software packages do exist. Most of them are specific to a single processing step or include different steps which are not specific to cloned DNA fragments analysis such as vector localization and removal. Besides, except the commercial Qiagen CLC Genomics workbench, there is no end-user oriented tool chaining all the steps from read assembly to annotation, providing an easily understandable report including result files. Therefore, we developed the MINTIA software package dedicated to automated analysis of sequences from cloned metagenomic DNA inserts. In this paper, we present MINTIA, results from simulated and real metagenomic data analysis and compare its results with three available open source software packages: fabFos (https://github.com/hallamlab/FabFos), Shims2 (*Bellott et al., 2018*) and RAST (*Aziz et al., 2008*).

## IMPLEMENTATION

### Description of the MINTIA program

MINTIA has been designed to process batches of functional metagenomic DNA fragments raw read files, usually sizing from 35 to 40 Kb, inserted into a fosmid (*e.g.*, PCC1FOS from Epicentre Technologies) in order to produce assembled and annotated contigs.

The 11 steps of the MINTIA workflow were grouped into two modules.

MINTIA Assemble module (Fig. 1) processes a compressed archive of fastq file(s) (single and/or paired reads) corresponding to distinct DNA inserts and the cloning vector sequence
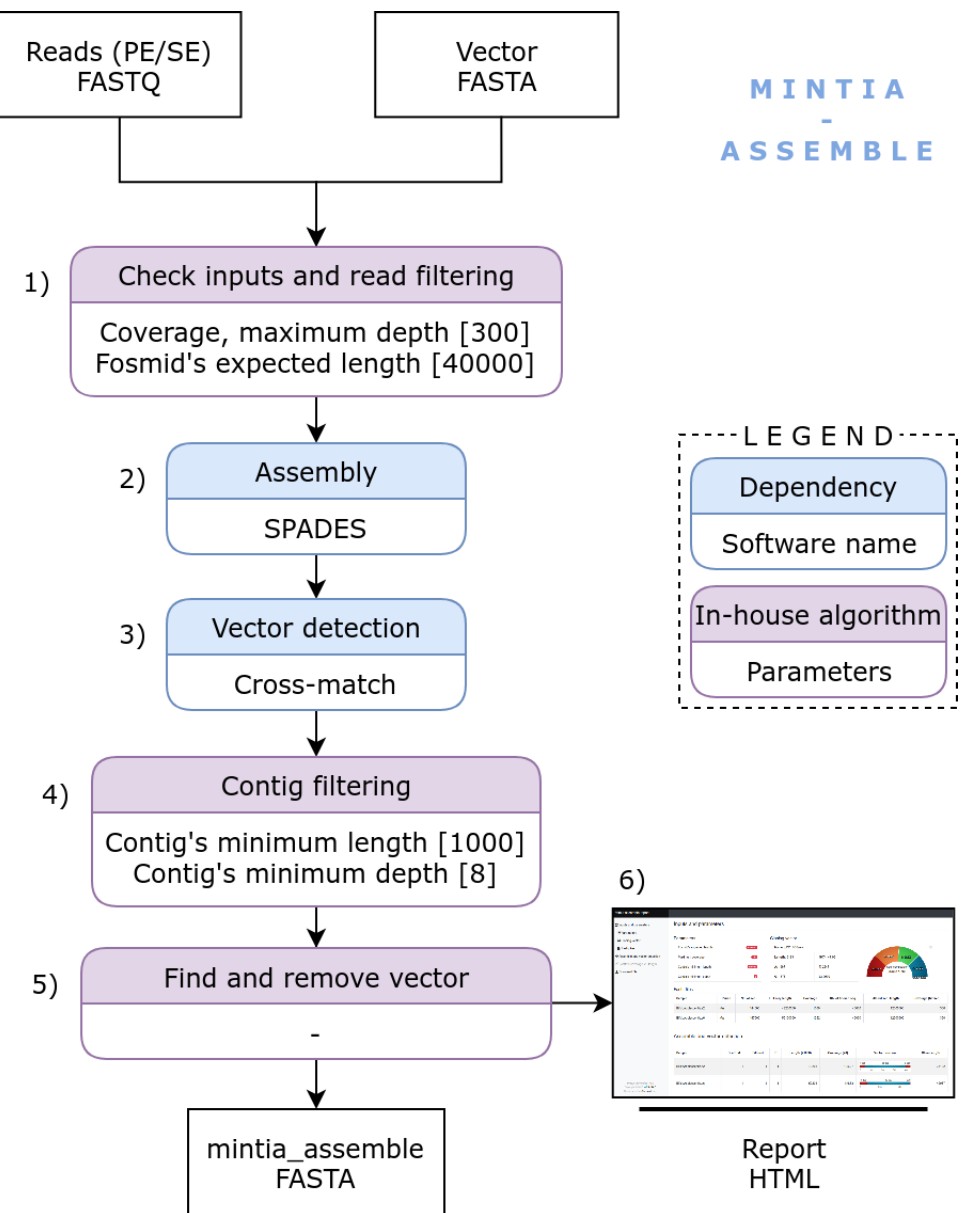

**Figure 1** MINTIA Assemble: The assembly module includes the following steps: (1) Input checking and read filtering, (2) Single and/or paired Fastq file(s) assembly using SPAdes, (3) Vector detection using Cross-match, (4) Contig filtering using an in-house script, (5) Vector removal, (6) Detailed and dynamic HTML report building.

in fasta format. In a preliminary step, read files are cleaned from adapters and reads of poor quality are filtered out using cutadapt (*Martin, 2011*). MINTIA Assemble first step counts reads and calculates read set coverage. Then a random read selection is performed to reduce each set to an appropriate user defined coverage (300X by default). Second, the assembly is performed with SPAdes (*Bankevich et al., 2012*) which produces raw contigs in fasta format. Third, the cloning vector is detected in the contigs and masked (with Xs) using

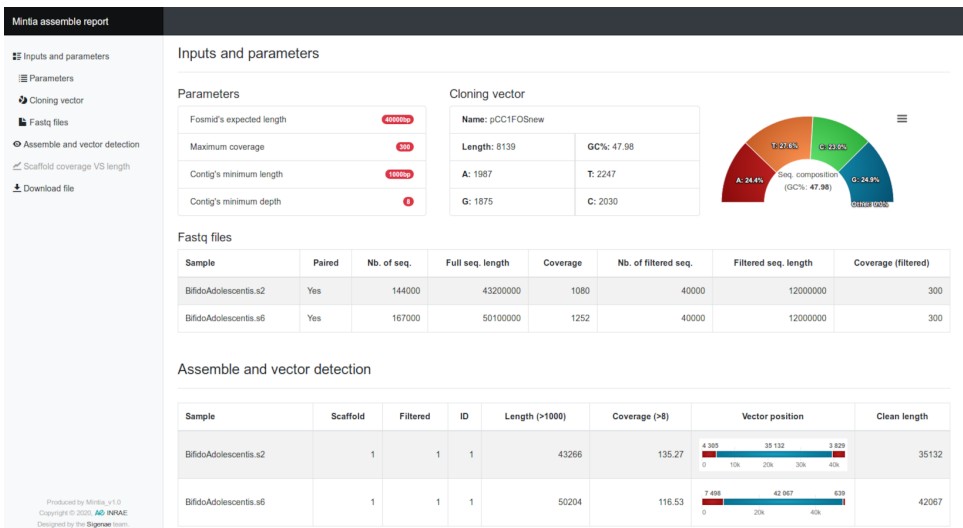

**Figure 2** **MINTIA assemble report.** (A) Main menu to navigate through the sections. (B) "Inputs and parameters" is a reminder of the parameters used as well as some statistics on the vector and the FASTQ files. (C) "Assemble and vector detection" is a table describing assembly results and vector location for each sample. The "Scaffold coverage *vs* length" (not shown) is a scatter plot where each dot presents coverage *versus* length of a contig. The "Download file" is a table including all the results in GFF and Fasta formats.

cross_match (*Ewing et al., 1998*). Fourth, contigs longer than 1000 base pairs with a mean coverage above eight are retained (minimal length and depth are parameters which can be modified). Fifth, the vector is removed and vector-cleaned contigs are re-organized to be consistent with fosmid circularity: if the vector is included in the contig, contig extremities are either joined if the extremities overlap each other or split into two smaller contigs if the extremities do not overlap. Last, a dynamic HTML report is produced (Fig. 2). It contains the input files description (reads and cloning vector) and the used parameters. It also contains a summary table and a graph which traces the different steps (assembly, vector cleaning and final contig building steps) in a user-friendly visual way. The report includes a table with links enabling to download the files generated in the different steps either for each initial DNA insert or for all DNA inserts as an archive. The HTML report file includes all graphics and download-able results files which makes it very easy to distribute. The mintia_assemble.fasta file contains contigs in fasta format and serves as the input of the annotation module.

The second module (MINTIA Annotate: Fig. 3) first detects CDS in contigs using prokka (*Seemann, 2014*). Then, CDS and contigs are aligned *versus* RefSeq non redundant (nr) (*OLeary et al., 2015*) and Uniprot/SwissProt (*The UniProt Consortium, 2016*) using diamond (*Buchfink, Xie & Huson, 2014*). It has been decided to align contigs (and not only CDS) *versus* nr and swissprot in order to be able to check if there were issues in the detection of CDS starting position. Results of these steps can be viewed in the HTML report as a table or in an igv.js based browser (*Robinson et al., 2020*). CDS are also aligned using rpsblast (*Camacho et al., 2009*) *versus* the COG database (*Tatusov et al., 2003*) and they can
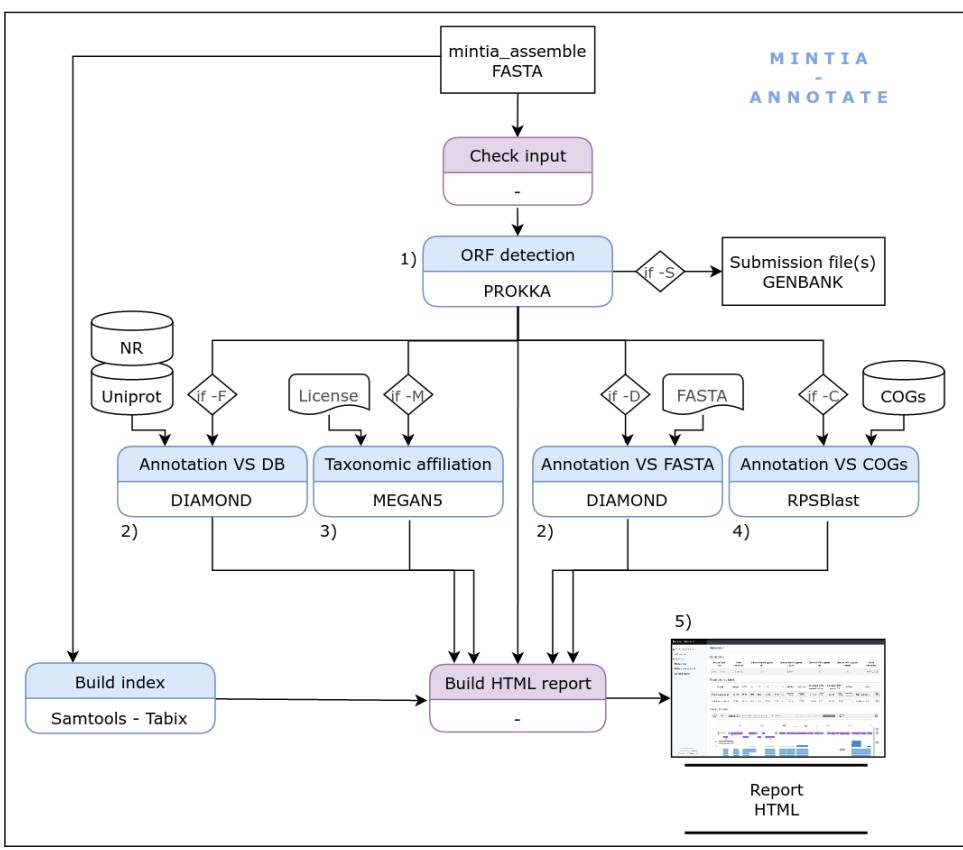

**Figure 3** MINTIA module 2: The annotate module includes following steps: (1) CDS detection using PROKKA, (2) Annotation against databases (NR, Uniprot/SwissProt, optional user provided Fasta file) with DIAMOND, (3) Taxonomic affiliation by MEGAN5, (4) Annotation against COG using RPSBlast, (5) Dynamic HTML report generation.

additionally be aligned with blast *versus* an in-house database provided by the user as a fasta file. MINTIA can also run MEGAN5 (*Huson et al., 2007*) in order to obtain taxonomic affiliations and can generate GENBANK compliant submission files. The Annotate module also produces a dynamic HTML report (Fig. 4) presenting the selected options and a table describing annotation results including links to individual or global result files.

## MINTIA program installation procedure

MINTIA installation comprises three steps. First, the software repository package is cloned from github (https://github.com/Bios4Biol/MINTIA). Second, a conda environment is created and populated with the software dependencies. MINTIA includes an install check module enabling to verify that all the needed software packages have been correctly installed. Third the reference annotation databases are downloaded and indexed.

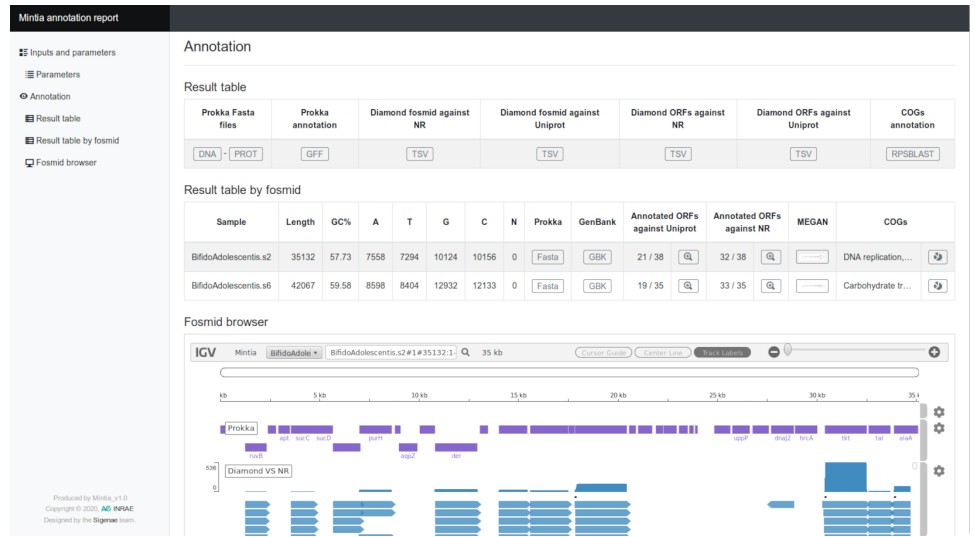

**Figure 4** **MINTIA annotate report.** (A) Main menu for section navigation. (B) Overall annotation result table allowing output file access. (C) Per fosmid annotation table comprising some statistics and allowing access to result files and graphics of the requested annotations. (D) igv.js based browser to explore the annotations (one track per annotation with for each glyph a contextual box with a link to external references).

## MATERIAL AND METHODS

### Simulations and biological examples
#### Simulations

First, 40,000 base pair long genomic chunks were randomly extracted from the genomes of *Bifidobacterium adolescentis ATCC 15703* (refered as Bifido later in the text - NCBI NC_008618.1 - complete genome of 2089645 bp), *Bacteroides stercoris ATCC 43183* scaffold 12 (refered as Sterco12 - NCBI NC_DS499673.1 - 587129 bp) and scaffold 16 (refered as Sterco16 - NCBI NZ_DS499677.1 - 468184 bp). Fifteen chunks were extracted for each genome.

Then the cloning vector (PCC1FOS of Epicentre Technologies) was inserted into those sequences, at a random position either at one of the extremities or inside the sequence. When the vector was inserted inside the sequence an overlap of 500 bp was created at the fosmid end to mimic circularity. This was performed using an in-house Python program.

Then, 150 base pair long paired reads (using art_illumina with parameters -ss HS25 -p -l 150 -f 1000 -m 200 -s 10) and 150 base pair long single reads (using art_454 with parameters -s and coverage 500) were generated for each sequence (called reference fosmid sequence later) using the ART software package (*Huang et al., 2011*). These read sets can be downloaded from https://doi.org/10.15454/TJC6SP.

fabFos, Shims2 and MINTIA were run on the simulated read sets in order to compare their assembly results. The number of contigs, sequence precision and recall were calculated to assess assembly quality. Precision and recall were calculated by aligning each assembled contig to the corresponding reference using minimap2 (*Li, 2018*) (version 2.7-r654) with

default parameters. The resulting alignment file was compressed, indexed and transformed into pileup format with samtools (*Li et al., 2009*) (version 0.1.19-96b5f2294a) with default parameters to count matching bases. The assembly precision corresponds to the ratio of the number of nucleotides correctly aligning to the reference over the contig length. The assembly recall corresponds to the number of correctly aligned nucleotides divided by the reference length. Mis-assemblies were manually checked to understand which genomic feature could explain them.

RAST and MINTIA were run on MINTIA assembled contigs to compare annotation results. RAST and MINTIA detected CDS were aligned with blat (*Kent, 2002*) version 36 using default parameters against the annotated CDS of the corresponding parts of the reference genome. True positive CDS are those aligning from start to end positions on their reference. The annotation precision corresponds to the ratio of CDS present in the genome detected by the software package over CDS present in the genome. The annotation recall corresponds to the ratio of CDS in the genome and detected by the software package over CDS detected by the software package. The quality and relevance of assembly and annotation were analyzed using R software package (version 4.0.2).

### Sequences from biological samples

In order to compare MINTIA results to a metagenomic DNA annotation performed by a biological expert, we processed fosmid reads produced in a functional metagenomic study aiming at identifying genes from mucin-degrading pathways originating from an uncultured fraction of the human gut microbiota (*Laville et al., 2019*). The reads corresponding to these biological samples can be downloaded from https://doi.org/10. 15454/AH8CRS.

## RESULTS

### Simulated samples

Fifteen reference fosmid sequences were randomly extracted for the three selected reference genomes. For each of the 45 fosmids, one MiSeq and one Proton read sets were simulated as described above. This resulted into 90 simulation sets, which were processed by fabFos, Shims2 and MINTIA.

As shown in Fig. 5A, among 90 simulated read sets, 24, 87 and 42 were assembled into a unique contig for fabFos, MINTIA and Shims2 respectively. fabFos produced 46, 17, 2 and 1 assemblies into 2, 3, 4 and 8 contigs respectively. Shims2 produced 2 and 1 assemblies comprising 2 and 4 contigs respectively. Shims2 not being able to assemble single reads these figures corresponded to 45 assemblies of paired reads. MINTIA produced 3 assemblies with 2 contigs.

Figures 5B and 5C present the precision and recall for the three pipelines on the same 90 simulations. Shims2 recall could not be calculated for single reads and is therefore not shown. The values of both measures have been grouped into four categories corresponding to the following intervals [99, 100], [90, 99), [70, 90) and [0, 70). fabFos precision values were all located in the third interval but five in the last interval. Shims2 precision values were all located in the first interval but one in the second. MINTIA precision values were all

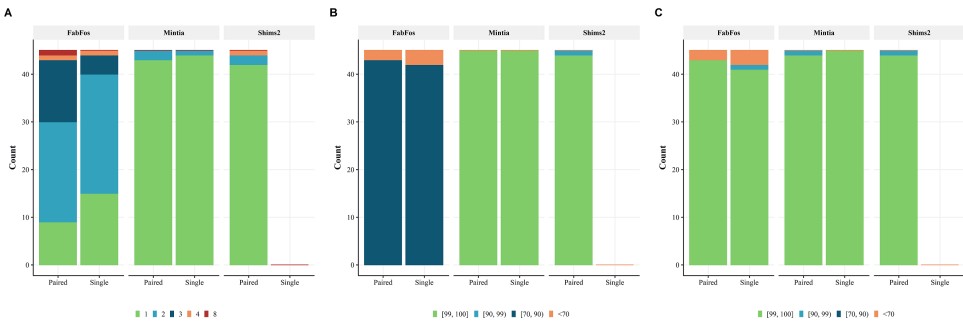

**Figure 5** **MINTIA assembly quality.** (A) The number of simulated sets assembled into 1, 2, 3, 4 or 8 contigs. (B) Assembly precisions grouped into four intervals [99, 100], [90, 99), [70, 90) and [0, 70). (C) Assembly recalls grouped into the same four intervals.

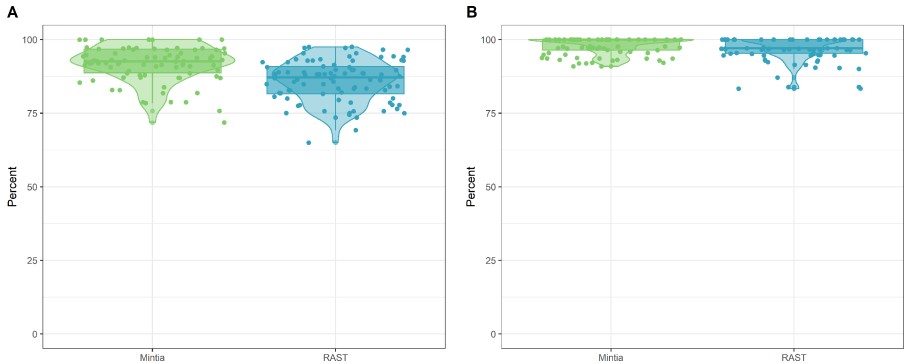

**Figure 6** **MINTIA annotation quality.** (A) The annotation precision of MINTIA in green and RAST in blue. (B) The annotation recall of MINTIA in green and RAST in blue.

located in the first interval. fabFos recall values were all but six in the first interval. Shims2 and MINTIA recall were all but one in the first interval.

The lowest recall in MINTIA assemblies was observed for the paired read assembly of Sterco16 simulation 11. This sequence contains a repeated region of 1,331 bp which splits up this fosmid into two assembled contigs and one part of contig 1 matches at both locations of this repeated region, explaining a difference in length between assembly and reference, leading to a decreased recall. Therefore the whole initial sequence is recovered but the repeated region is present only once in the assembled contigs.

RAST and MINTIA annotation qualities were compared to each other on the same set of contigs. For each contig the precision and recall were calculated and presented in Fig. 6. RAST has a mean (standard deviation) for precision of 85.89 (7.08) and a mean recall of 96.86 (4.18). MINTIA has a mean precision of 92.07 (6.20) and a mean recall of 97.92 (2.86).

### Biological samples

Thirteen fosmids sequences suspected to contain genes coding for mucin-degrading enzymes were assembled and annotated with MINTIA and compared to previously published results (*Laville et al., 2019*).

These fosmids were all assembled in one contig as in the published assemblies. The MINTIA assembled contigs had nearly the same length than the reference contigs (difference in length is of 0 in 2 cases, of +1 in 2 cases, of +2 in 8 cases and of +5 in one case). Aligning MINTIA contigs *versus* reference contigs with blat version 36 with default parameters led to identity percentages higher than 99.9% and to query coverage of 100%.

In eight cases out of thirteen, the number of CDS detected by MINTIA matched the number of CDS found by manual expertise. MINTIA detected one more CDS in four cases and three CDS less in one case.

## DISCUSSION

In order to evaluate MINTIA assembly results, we compared them to those of fabFos and Shims2. fabFos produced much more contigs than Shims2 and MINTIA because it does not include circularization and filtering steps and therefore produced several contigs where both other software packages only generated one contig as expected. fabFos precision is much lower than both others because it does did not produce vector cleaned contigs. fabFos includes a vector search step but does not output a vector cleaned contig fasta file which is the classical input for the annotation step. fabFos recall was close to the ones observed for the two other software packages but still lower. This could originate from the larger kmer size used in the SPAdes assembly step. Shims2 produced mainly one contig per assembly but only for paired reads. It still had a few more assemblies with several contigs than MINTIA for paired reads. Shims2 precision and recall were very high for paired reads and very similar to the ones observed for MINTIA. Overall on this simulated reads sets MINTIA assemble performed best regarding the three measured criteria.

In order to evaluate MINTIA annotation results, we compared them to RAST results on the same contig set. RAST is a well-known frequently used prokaryote sequence annotation pipeline. Specificity and recall were calculated after aligning the CDS sequences produced by both pipelines to the ones of the reference annotation. MINTIA showed a better precision and recall mean than RAST with a smaller standard deviation.

The MINTIA pipeline was developed to be easy to use by biologists: two command lines are enough to run eleven steps performing a complete analysis from read filtering to submission file generation. MINTIA allows to quickly run assembly and first annotation steps on a large set of fosmids and its results are displayed in a graphical WEB interface which is easy to scroll in order to see the different results. For further exploration MINTIA allows to blast in a very easy way the assembled fosmids against any in-house database by just providing the reference fasta file. MINTIA is a free and open-source software package.

## CONCLUSION

In this paper, we described MINTIA, a pipeline for fosmid reads assembly and annotation which can be run on a *nix workstation or server. It allows researchers to perform functional annotation of vector cloned metagenomic inserts. MINTIA links different previously existing software packages in a proper order and provides results as HTML pages including graphical views as well as links to all the intermediate and final output files. MINTIA assemble module outperformed fabFos and Shims2 on a large set of simulated metagenomic reads regarding contig count, precision and recall. Its annotation module also showed better precision and recall figures than RAST on part of this set.

## ACKNOWLEDGEMENTS

We are grateful to the Genotoul bioinformatics platform Toulouse Occitanie (Bioinfo Genotoul, doi: 10.15454/1.5572369328961167E12) for providing help, computing and storage resources.

### Funding
The authors did not receive external funding.

### Competing Interests
The authors declare there are no competing interests.

### Author Contributions
- Philippe Bardou performed the experiments, analyzed the data, prepared figures and/or tables, authored or reviewed drafts of the paper, and approved the final draft.
- Sandrine Laguerre conceived and designed the experiments, performed the experiments, analyzed the data, prepared figures and/or tables, authored or reviewed drafts of the paper, and approved the final draft.
- Sarah Maman Haddad and Sabrina Legoueix Rodriguez performed the experiments, authored or reviewed drafts of the paper, and approved the final draft.
- Elisabeth Laville and Christophe Klopp conceived and designed the experiments, performed the experiments, authored or reviewed drafts of the paper, and approved the final draft.
- Claire Dumon and Gabrielle Potocki-Veronese conceived and designed the experiments, authored or reviewed drafts of the paper, and approved the final draft.

### Data Availability
The software package is available at GitHub: https://github.com/Bios4Biol/MINTIA.

The simulated reads sets are available at the INRAE network: Laguerre, Sandrine, 2020, "MINTIA paper Simulated data", https://doi.org/10.15454/TJC6SP, Portail Data INRAE, V1.

The reads corresponding to the biological samples are available at the INRAE network: Laguerre, Sandrine, 2020, "MINTIA paper Biological samples", https://doi.org/10.15454/AH8CRS, Portail Data INRAE, V1.

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
