# Peer review of "MINTIA: a metagenomic INserT integrated assembly and annotation tool"

_PeerJ, doi:10.7717/peerj.11885_

## Round 0.1 · original submission · Major Revisions

Dear Dr. Bardou and colleagues:

Thanks for submitting your manuscript to PeerJ. I have now received three independent reviews of your work, and as you will see, one reviewer recommended rejection, while the others suggested minor revisions (though with many suggested changes). I am affording you the option of revising your manuscript according to all reviews but understand that your resubmission may be sent to at least one new reviewer for a fresh assessment (unless the reviewer recommending rejection is willing to re-review).

All three reviewers find that MINTIA, which enables biologists to analyze metagenomic data retrieved after activity-based screening, generates friendly reports and files ready for submission after a series of user-applied step. However, your research question is not well defined in the manuscript. Nearly all results in this manuscript are just brief descriptions, as there are no figures or tables to support your opinions/observations. Furthermore, and probably the biggest shortcoming, is that there are no comparisons of MINTIA and similar available pipelines. What are the advantages of MINTIA over other approaches?

There seem to be missing references regarding fosmids/cosmids, as well as an effective approach for handling large sequences.

There are many minor suggestions to improve the manuscript. Importantly, please ensure that an English expert has edited your revised manuscript for content and clarity.

Therefore, I am recommending that you revise your manuscript, accordingly, taking into account all of the issues raised by the reviewers.

Good luck with your revision,

-joe

Reviewer 1 ·

Basic reporting

no comment

Experimental design

no comment

Validity of the findings

no comment

Additional comments

Bardou et al. present in this manuscript a tool providing an end-to-end pipeline to assemble and analyze functional metagenomics data. The tool (MINTIA) relies on a number of established “standard” utilities, and provide a user-friendly and intuitive interface so that biology researchers can easily process their fosmid read data.
Overall, the concept of the tool is sound, and the interface well designed. My only concern is the reliance on “cross_match” from the Phrap/cross_match/swat package, which requires a specific license. I would encourage the authors to look for open source / free-to-use alternatives to this program for future versions of their tool.
Other than that, I only have minor comments listed below:

l. 24: “easily generate and analyze large”: To my understanding, MINTIA does not provide a way to “generate” fosmid data, so this should probably be modified to e.g. “easily assemble and analyze … ” ?

l. 72: “read files are cleaned from adaptors and reads of poor quality are filtered”: Could the authors specify here whether this is done using existing tools or via an in-house script ? Same question for the read selection step (l. 73)

l. 90: “CDS and contigs are aligned using diamond”: Are the nucleotide contigs directly compared to nr and Uniprot/Swissprot ? If so, I believe this does not appear on Fig. 3, and I would encourage the authors to mention to the reader why the contigs are also used here, and not just the predicted cds.

l. 117: “generared” should be “generated”

l. 124: should “standard parameters” be “default parameters” ?

l. 135: “each of the 3 selected reference genomes”: Please specify the 3 genome/contig IDs here.

l. 166: “CDS, had” should be “CDS had”

l. 221: “we described” should be “we describe”

Figure 2: Should there be an indication (symbol ?) somewhere on this page to indicate whether a contig is circular or not ? Or maybe on the view shown in Fig. 4 ? (Apologies if it already there and I missed it).

Figure 3: The legend and figure mention “Functional classification” using MEGAN, however the main text indicates instead “MEGAN5 ( Huson DH (2007)) in order to obtain taxonomic affiliations” (l. 95), and to my knowledge MEGAN is indeed a tool for taxonomic, not functional, annotation. Please clarify

Figure 5: “Figure A” should be “Panel A”, and “Figure B” should be “Panel B”

Reviewer 2 ·

Basic reporting

1. The study suffers greatly from the use of English. I cannot make a final assessment of the study unless English is proficient.

2. The reference of “Tauzin et al. 2020” is missing and format is wrong (Page 1, line 42-43). And literature references is not sufficient. Please check all references and add references in introduction and discussion sections. For example, RAST server has no reference (Page 7, line 191).

3. Some results in this manuscript are just brief description. It has no figures or tables to support author’s opinions.

For example, “Moreover MINTIA …” (Page 5, line 152-156). In the paragraph “The lowest precision…” (Page 6, line 156-162),please add figures or tables to exhibit these results.

In result sections, most of sentences describe data results, but lacking summarized sentences to exhibit the meaning of these results.

Experimental design

There are two main flaws in the current version of this manuscript .

The first one is lacking of comparisons with already available assembly pipelines, fabFos and shims2, in coverage, length and other results using the same data, In the same, lacking of comparison with orient tools, QIAGEN CLC genomics workbench, in annotated data and other results. Which step is the difference between MINTIA and these tools, and improve what? What is the difference in sensitivity and precision between MINTIA and these tools?

The second one is that, except friendly reports and files ready for submission, MINTIA have other advantages or not. If have, please add them in abstract and introduction sections. If not, it is not enough to exhibit the advantages of MINTIA.

Validity of the findings

In Figure 2 legend, the scatter plot “Scaffold coverage VS length” (not show) should add in supplementary figure.

Additional comments

1. The first letter of each word in the current title “MINTIA: a Metagenomic INserT Integrated Assembly and annotation tool” should be unified. For example, “MINTIA: A Metagenomic INserT Integrated Assembly and Annotation Tool” or “MINTIA: a metagenomic insert integrated assembly and annotation tool”.

2.Don’t use oral words or words with emotion. For example, “thanks to” (Page 1, Line 31), “Everyday” (Page 1, Line 34), “Careful analysis” (Page 2, line 53-54), Please check them in all manuscript.

3. “Its throughput, cost and versatility have been continuously improved in the last decade, in particular thanks to droplet microfluidics (Tauzin et al. 2020).” Its refer what? Cost improving is good thing or bad thing? (Page 1, Line 41-43)

4. How to transform a microbial expression host, please describe clearly. (Page 2, Line 45)

5. “NGS technologies…”, add complete spellings on first occurrence. Please check all manuscript. (Page 2, Line 49)

6. “Genes, their genomic context …”, it is too long to understand this sentence. (Page 2, Line50)

7. Read files are cleaned after filtering adapter and reads of poor quality. (Page 2, Line 72)

8. “Align with blast versus an in-house database provided …”, software can’t add behind “align with”. Moreover, please add the version of the software you used. (Page 2, Line 94)

9. “When the vector …”, this sentence is ungrammatical, please revise it. (Page 4, line 115-116)

10. “Sensitivity corresponds …”, these sentences are too hard to understand. Please revise them. (Page 4 ,line 124-127)

11. “17 had a difference …”, what is the subject,“17”and “3”? (Page 5, line 141-142)

12. Please add specific results in comparing MINTIA and the existing tools and exhibit them using figures and tables. (Page 6-7, Line 180-198)

Reviewer 3 ·

Basic reporting

For context, it might be helpful to cite the work of Lam et al. https://doi.org/10.1371/journal.pone.0098968 who addressed challenges in the sequencing of cosmid clones by using a pooled strategy.

Experimental design

No comment

Validity of the findings

No comment

Additional comments

Several places where the word "chains" could be replaced with the word "links"
Line 49: change "accessed" to "assessed"
Line 117: change "generared" to "generated"
Line 129: Unclear what the meaning of "expertized" is
Line 131: A brief description of the sequencing parameters would be helpful
Line 222: Change "Unix" to "*nix"

---

## Round 0.2 · accepted · Accept

Dear Dr. Bardou and colleagues:

Thanks for revising your manuscript based on the concerns raised by the reviewers. I now believe that your manuscript is suitable for publication. Congratulations! I look forward to seeing this work in print, and I anticipate it being an important resource for groups studying annotation and assembly approaches for metagenomics data. Thanks again for choosing PeerJ to publish such important work.

Best,

-joe

Reviewer 1 ·

Basic reporting

N/A

Experimental design

N/A

Validity of the findings

N/A

Additional comments

I thank the authors for addressing the different comments, and have no further question or suggestion at this point.

Reviewer 3 ·

Basic reporting

N/A

Experimental design

N/A

Validity of the findings

N/A

Additional comments

The authors have addressed my concerns to my satisfaction.